# A Pooled Analysis of Preoperative Inflammatory Biomarkers to Predict 90-Day Outcomes in Patients with an Aneurysmal Subarachnoid Hemorrhage: A Single-Center Retrospective Study

**DOI:** 10.3390/brainsci13020257

**Published:** 2023-02-02

**Authors:** Zhaobo Nie, Fa Lin, Runting Li, Xiaolin Chen, Yuanli Zhao

**Affiliations:** 1Department of Neurosurgery, Beijing Tiantan Hospital, Capital Medical University, Beijing 100070, China; 2Department of Neurosurgery, Beijing Shunyi Hospital, Capital Medical University, Beijing 101300, China; 3China National Clinical Research Center for Neurological Diseases, Beijing 100070, China; 4Center of Stroke, Beijing Institute for Brain Disorders, Beijing 100070, China; 5Beijing Key Laboratory of Translational Medicine for Cerebrovascular Disease, Beijing 100070, China

**Keywords:** aneurysmal subarachnoid hemorrhage, inflammatory biomarkers, pneumonia, functional outcome

## Abstract

An inflammatory response after an aneurysmal subarachnoid hemorrhage (aSAH) has always been in the spotlight. However, few studies have compared the prognostic impact of inflammatory biomarkers. Moreover, why these inflammatory biomarkers contribute to a poor prognosis is also unclear. We retrospectively reviewed aSAH patients admitted to our institution between January 2015 and December 2020. The 90-day unfavorable functional outcome was defined as a modified Rankin scale (mRS) of ≥ 3. Independent inflammatory biomarker-related risk factors associated with 90-day unfavorable outcomes were derived from a forward stepwise multivariate analysis. Receiver operating characteristic curve analysis was conducted to identify the best cut-off value of inflammatory biomarkers. Then, patients were divided into two groups according to each biomarker’s cut-off value. To eliminate the imbalances in baseline characteristics, propensity score matching (PSM) was carried out to assess the impact of each biomarker on in-hospital complications. A total of 543 patients were enrolled in this study and 96 (17.7%) patients had unfavorable 90-day outcomes. A multivariate analysis showed that the white blood cell (WBC) count, the systemic inflammation response index, the neutrophil count, the neutrophil-to-albumin ratio, the monocyte count, and the monocyte-to-lymphocyte ratio were independently associated with 90-day unfavorable outcomes. The WBC count showed the best predictive ability (area under the curve (AUC) = 0.710, 95% CI = 0.652–0.769, *p* < 0.001). After PSM, almost all abnormal levels of inflammatory biomarkers were associated with a higher incidence of pneumonia during hospitalization. The WBC count had the strongest association with poor outcomes. Similar to nearly all other inflammatory biomarkers, the cause of poor prognosis may be the higher incidence of in-hospital pneumonia.

## 1. Introduction

Aneurysmal subarachnoid hemorrhages (aSAH) are one of the most common neurological emergencies with a high mortality rate of 22–50%, and even patients who receive optimal medical care may experience a long-term disability and cognitive impairment [1,2]. There are many reasons for this unfavorable outcome, of which researchers generally believe that early brain injuries (EBI) may be the most important cause of poor patient prognoses. EBI is a complex concept used to explain the pathophysiological events in the brain within 72 h after aneurysm rupture, including 0energy failure, ionic changes, increased endothelin-1 levels, depletion of nitric oxide, inflammation, oxidative stress, etc. [3,4,5,6]. Among these, inflammation after a hemorrhage is an important factor affecting the outcomes of aSAH patients, since some changes are associated with the development of cerebral vasospasm (CVS) and delayed cerebral ischemia (DCI) [7,8,9,10].

Recently, many studies have revealed the relationship between inflammatory biomarkers and a poor prognosis of patients with aSAH, such as white blood cells (WBC), neutrophils (NEUT), and monocytes (MONO) [11,12,13]. In addition, some derived inflammatory biomarkers have already been reported to be associated with unfavorable outcomes in patients with aSAH or ischemic stroke, including the neutrophil-to-lymphocyte ratio (NLR), the neutrophil-to-albumin ratio (NAR), the lymphocyte-to-monocyte ratio (LMR), the monocyte-to-lymphocyte ratio (MLR), the platelet-to-white blood cell ratio (PWR), the platelet-to-neutrophil ratio (PNR), the platelet-to-lymphocyte ratio (PLR), the mean platelet volume-to-platelet count ratio (MPV/PLT), the systemic inflammation response index (SIRI, neutrophil count × monocyte count/lymphocyte count), and the systemic immune-inflammation index (SII, platelet count × neutrophil count/lymphocyte count) [14,15,16,17,18,19,20,21]. Although they all perform well in predicting poor outcomes according to the previous studies, unfortunately, most of these biomarkers have been reported separately, which means few studies have compared their predictive ability together in aSAH patients. Moreover, the reasons that these biomarkers lead to poor prognoses have also not been widely recognized. Finding out the best inflammatory biomarker and exploring the causes of poor prognosis have great significance, which can help clinicians better understand the role of the inflammatory response in the prognoses of patients with aSAH.

Therefore, the objectives of this study were to analyze the associations between inflammatory biomarkers and outcomes of aSAH patients, compare the ability of these inflammatory biomarkers to predict outcomes, and try to find out potential reasons for these inflammatory biomarkers leading to unfavorable outcomes.

## 2. Materials and Methods

### 2.1. Study Design

We retrospectively analyzed the clinical data of aSAH patients hospitalized in the neurosurgery department of Beijing Tiantan Hospital between January 2015 and December 2020. All patients’ data were derived from the Long-term Prognosis of Emergency Aneurysmal Subarachnoid Hemorrhage (LongTEAM) Registry study (Registration No. NCT04785976). In the present study, all patients had angiographically documented aneurysms with subarachnoid hemorrhage (SAH), confirmed by either computed tomography (CT) or lumbar puncture. The inclusion criteria were as follows: (1) age ≥ 18 years old; (2) emergency admission; (3) less than 72 h from the rupture to the admission and less than 72 h from the admission to treatment; (4) a single aneurysm; and (5) treated with surgical clipping or endovascular coiling. The exclusion criteria were as follows: (1) previous SAH; (2) history of neurosurgery due to any cause; (3) physical disability due to any previous disease; and (4) insufficient medical, radiological, and laboratory information.

This study was approved by the Institutional Review Board of Beijing Tiantan Hospital. Informed consent for clinical analysis was obtained from all individual participants or their authorized representatives. All the studies were performed according to the Declaration of Helsinki and the local ethics policies. All patients were managed according to the guidelines [22].

### 2.2. Patients’ Data

From patients’ medical records, we collected their (1) demographic information and their medical history, including age, gender, hypertension, and diabetes mellitus; (2) lifestyle risk factors, including current smoking history; (3) location of aneurysm; (4) first CT-related information on admission, including their modified Fisher scale (mFS) grade, Graeb score, Subarachnoid Hemorrhage Early Brain Edema Score (SEBES) [23], and acute hydrocephalus; (5) World Federation of Neurological Societies (WFNS) grade on admission; (6) first inflammation-related laboratory examination on admission, including their WBC count (normal range: (3.5–9.5) × 10^9^/L), MONO count (normal range: (0.10–0.60) × 10^9^/L), lymphocyte count (LY, normal range: (1.10–3.2) × 10^9^/L), NEUT count (normal range: (1.80–6.30) × 10^9^/L), PLT count (normal range: (125–350) × 10^9^/L), NLR, NAR, SIRI, SII, MLR, LMR, PWR, PLR, MPV/PLT, and PNR; (7) treatment modality, including surgical clipping and endovascular coiling; and (8) in-hospital complications, including DCI, intracranial infection, stress ulcer bleeding, hypoproteinemia, pneumonia, and deep vein thrombosis.

### 2.3. Outcome Measurement

The primary outcome was the functional neurological outcome, measured by the modified Rankin scale (mRS) 90 days after discharge (the neurosurgeon followed up with patients via telephone or an outpatient appointment). Patients were divided into an unfavorable outcome group with mRS ≥ 3. The second outcome was the occurrence of in-hospital complications.

### 2.4. Statistical Analysis

The analysis was performed with SPSS Statistics 26.0 (IBM, Armonk, New York, NY, USA) and GraphPad PRISM 8.3.0. (GraphPad Software Inc., San Diego, CA, USA). Significance was set at *p* < 0.05.

Continuous variables were summarized as means ± standard deviation or medians with interquartile range (IQR). Categorical variables were summarized as frequency (percentage). After testing for normality, continuous variables were analyzed using the independent Student *t*-test (normal distribution), the Mann–Whitney U test or the Kruskal–Wallis H test. The Pearson chi-square test or Fisher’s exact test were performed to test the dichotomized variables. Due to the lack of daily biomarkers data, the point at which each person’s blood was first drawn was included in the analysis. The time the blood sample of inflammatory biomarkers was taken was classified into three groups (Day 1 (rupture to admission interval between 0 and 24 h), Day 2 (rupture to admission interval between 25 and 48 h), and Day 3 (rupture to admission interval between 49 and 72 h)) to find out whether there were different levels in the predefined groups over time.

Only variables with *p* < 0.05 in univariate analysis were entered in a forward stepwise likelihood ratio multivariate logistic model to identify the inflammation-related independent risk factors associated with 90-day unfavorable outcomes. The odds ratios (OR) and 95% confidence intervals (CI) of variables were calculated. Interaction terms were used to investigate whether the association between inflammatory biomarkers and the 90-day unfavorable outcomes differed according to the WFNS grade group or treatment modality. Afterward, subgroup analysis was conducted in the same manner as for the primary outcome in patients in the same WFNS grade and treatment modality group.

Receiver operating characteristic (ROC) analysis was performed to evaluate associations between the inflammatory biomarkers on the admission and 90-day outcomes to determine the best cut-off value for the prediction. Furthermore, we used the area under the curve (AUC) in ROC analysis to compare the predictive ability of different inflammatory biomarkers. To observe the impact of inflammatory biomarkers on in-hospital complications, we performed propensity score matching (PSM) with a match tolerance of 0.02 and a ratio of 1:1 to adjust for imbalances of baseline characteristics such as age, hypertension, WFNS grade, mFS grade, Graeb score, acute hydrocephalus, and surgical clipping between the two outcome groups. After PSM, according to each inflammatory biomarker’s cut-off value, we divided the patients into two groups to observe the relationship between various inflammatory biomarkers and in-hospital complications.

## 3. Results

A total of 543 patients were enrolled in this study. A total of 96 (17.7%) patients had unfavorable 90-day outcomes and 447 (82.3%) patients had favorable 90-day outcomes. Compared with patients with favorable outcomes, patients who had unfavorable 90-day outcomes were older (58.7 ± 10.8 vs. 53.9 ± 11.0, *p* < 0.001), more likely to choose surgical clipping, had a higher incidence of hypertension history (71/96 (74.0%) vs. 255/447 (57.0), *p* = 0.002), had a higher proportion of WFNS grade 4–5 on admission (56/96 (58.3%) vs. 73/447 (16.3%), *p* < 0.001), had a higher proportion of mFS grade 3–4 on admission (89/96 (92.7%) vs. 339/447 (75.8%), *p* < 0.001), had a higher proportion of Graeb score 4–5 on admission (22/96 (22.9%) vs. 26/447 (5.8%), *p* < 0.001), and had a higher proportion of acute hydrocephalus on admission (50/96 (52.1%) vs. 175/447 (39.1%), *p* = 0.020). In addition, patients who had 90-day unfavorable outcomes had a significantly higher WBC count (median (IQR) 15.26 (12.44–18.54) vs. 11.86 (9.40–14.68), *p* < 0.001), MONO count (median (IQR) 0.57 (0.40–0.81) vs. 0.39 (0.26–0.54), *p* < 0.001), NEUT count (median (IQR) 13.56 (11.03–16.34) vs. 10.51 (7.97–13.27), *p* < 0.001), NLR (median (IQR) 15.21 (8.99–21.00) vs. 11.06 (6.97–15.70), *p* < 0.001), NAR (median (IQR) 0.32 (0.25–0.39) vs. 0.25 (0.19–0.31), *p* < 0.001), SIRI (median (IQR) 7.06 (4.13–11.70) vs. 3.68 (2.36–6.24), *p* < 0.001), SII (median (IQR) 3412 (2089–5002) vs. 2470 (1535–3548), *p* < 0.001), and MLR (median (IQR) 0.57 (0.36–0.79) vs. 0.37 (0.26–0.54), *p* < 0.001), but lower LMR (median (IQR) 1.77 (1.26–2.79) vs. 2.70 (1.87–3.79), *p* < 0.001), PWR (median (IQR) 15.26 (12.13–19.80) vs. 19.00 (14.96–23.51), *p* < 0.001), and PNR (median (IQR) 17.67 (13.29–21.10) vs. 21.69 (16.60–27.92), *p* < 0.001) (Table 1).

After the rupture event occurred, the levels of inflammatory biomarkers were analyzed. A total of 298 patients (54.9%) received tests on post-hemorrhagic day 1, 158 (29.1%) on day 2, and 87 (16.0%) on day 3. The levels of inflammatory biomarkers according to three blood sample timings are shown in Figure 1. On Day 1, Day 2, and Day 3, the levels of 10, 7, and 12 inflammatory biomarkers, respectively, were different across the 90-day unfavorable outcomes.

*P*_1_ was determined by the Kruskal–Wallis H test for different levels of inflammatory biomarkers in three groups. *P*_2_ was determined by the Mann–Whitney U test for different levels of inflammatory biomarkers between 90-day favorable and unfavorable outcome groups. *P*_day1_, *P*_day2_, and *P*_day3_ were determined by the Mann–Whitney U test for different levels of inflammatory biomarkers according to three groups between 90-day favorable and unfavorable outcome groups.

### 3.1. Inflammatory Biomarker-Related Risk Factors Associated with 90-Day Unfavorable Outcomes

The inflammatory biomarkers of significance (*p* < 0.05) in univariate analysis were successively analyzed by multivariate analysis. After adjusting for age, hypertension, WFNS grade 4–5, mFS grade 3–4, a Graeb score of 5–12, acute hydrocephalus, and treatment modality, the multivariate analysis showed that the WBC count (OR = 1.15, 95% CI = 1.08–1.22, *p* < 0.001), SIRI (OR = 1.09, 95% CI = 1.04–1.14, *p* < 0.001), NEUT count (OR = 1.14, 95% CI = 1.08–1.22, *p* < 0.001), NAR (OR = 110.19, 95% CI = 9.42–1288.36, *p* < 0.001), MONO count (OR = 7.38, 95% CI = 2.75–19.76, *p* < 0.001), and MLR (OR = 4.70, 95% CI = 1.92–11.48, *p* = 0.001) were independently associated with 90-day unfavorable outcomes in aSAH patients (Table 2). In addition, the results of subgroup analyses of the WFNS grade and treatment modality on the primary outcome are shown in Table 3 and Table 4. In a subgroup analysis of the two WFNS grades, there are interactions between inflammatory biomarkers (i.e., WBC, NEUT, NAR, and MONO) and 90-day unfavorable outcomes. In a subgroup analysis of the two treatment mortality, there are interactions between inflammatory biomarkers (i.e., WBC, SIRI, NEUT, NAR, MONO, and MLR) and 90-day unfavorable outcomes.

### 3.2. Associations between Inflammatory Biomarkers and WFNS Grade, mFS Grade, and Graeb Score

Except for LMR, PWR, and PNR, which showed lower levels for worse clinical grades, all other biomarkers showed higher levels for worse clinical grades (Table 5).

### 3.3. Receiver Operating Characteristic Curve Analysis

Among all inflammatory biomarkers, the three which most strongly associated with unfavorable outcomes were the WBC count (AUC = 0.710, 95% CI = 0.652–0.769, *p* < 0.001), SIRI (AUC = 0.708, 95% CI = 0.650–0.767, *p* < 0.001), and the NEUT count (AUC = 0.701, 95% CI = 0.642–0.760, *p* < 0.001) (Figure 2). ROC analysis identified a WBC count of 14.82 × 10^9^/L (sensitivity = 49%, specificity = 76%, and Youden’s index = 0.35), SIRI of 6.77 (sensitivity = 55%, specificity = 78%, and Youden’s index = 0.34), NEUT count of 11.39 × 10^9^/L (sensitivity = 73%, specificity = 61%, and Youden’s index = 0.34), NAR of 0.29 (sensitivity = 64%, specificity = 70%, and Youden’s index = 0.33), MONO count of 0.55 × 10^9^/L (sensitivity = 55%, specificity = 77%, and Youden’s index = 0.32), LMR of 1.79 (sensitivity = 52%, specificity = 77%, and Youden’s index = 0.29), MLR of 0.56 (sensitivity = 51%, specificity = 78%, and Youden’s index = 0.29), PWR of 15.62 (sensitivity = 54%, specificity = 71%, and Youden’s index = 0.25), PNR of 20.72 (sensitivity = 69%, specificity = 56%, and Youden’s index = 0.24), SII of 3102 (sensitivity = 58%, specificity = 67%, and Youden’s index = 0.25), and NLR of 14.88 (sensitivity = 52%, specificity = 71%, and Youden’s index = 0.23) as the best cut-off values to discriminate favorable or unfavorable 90-day outcomes in aSAH patients.

### 3.4. Associations between Inflammatory Biomarkers and In-Hospital Complications

We adjusted for imbalances of baseline characteristics such as age, hypertension, WFNS grade, mFS grade, Graeb score, acute hydrocephalus, and surgical clipping. There was no statistical difference in baseline characteristics between the two groups after PSM (Table 6).

According to each inflammatory biomarker’s cut-off value, patients with a WBC count of >14.82 × 10^9^/L had a higher incidence of in-hospital pneumonia (55/80 (68.8) vs. 38/92 (41.3%), *p* < 0.001) when compared with patients with a WBC count of ≤14.82 × 10^9^/L. Patients with a SIRI of >6.77 had a higher incidence of in-hospital pneumonia (49/74 (66.2%) vs. 44/98 (44.9%), *p* = 0.006) when compared with patients with a SIRI of ≤6.77. Patients with an NEUT count of >11.39 × 10^9^/L had a higher incidence of in-hospital pneumonia (63/100 (63.0%) vs. 30/72 (41.7%), *p* = 0.006) when compared with patients with an NEUT count of ≤11.39 × 109/L. Patients with an NAR of >0.29 had a higher incidence of in-hospital pneumonia (59/86 (68.6%) vs. 34/86 (39.5%), *p* < 0.001) when compared with patients with an NAR of ≤0.29. Patients with a MONO count of >0.55 × 10^9^/L had a higher incidence of in-hospital intracranial infection (13/67 (19.4%) vs. 9/105 (8.6%), *p* < 0.001) when compared with patients with a MONO count of ≤0.55 × 10^9^/L. Patients with an LMR of <1.79 had a higher incidence of in-hospital pneumonia (47/71 (66.2%) vs. 46/101 (45.5%), *p* = 0.008) when compared with patients with an LMR of >1.79. Patients with an MLR of >0.56 had a higher incidence of in-hospital pneumonia (45/69 (65.2%) vs. 48/103 (46.6%), *p* = 0.016) when compared with patients with an MLR of ≤0.56. Patients with a PWR of <15.62 had a higher incidence of in-hospital pneumonia (53/80 (66.3%) vs. 40/92 (43.5%), *p* = 0.003) when compared with patients with a PWR of ≥15.62. Patients with a PNR of <20.72 had a higher incidence of in-hospital pneumonia (64/102 (62.7%) vs. 29/70 (41.4%), *p* = 0.006) when compared with patients with a PNR of ≥20.72. Patients with an SII of >3102 had a higher incidence of in-hospital pneumonia (55/84 (65.5%) vs. 38/88 (43.2%), *p* = 0.003) when compared with patients with an SII of ≤3102. Patients with an NLR of >14.88 had a higher incidence of in-hospital pneumonia (47/74 (63.5%) vs. 46/98 (46.9%), *p* = 0.031) when compared with patients with an NLR of ≤14.88 (Table 7).

## 4. Discussion

This large retrospective study reviewed several accessible clinical inflammatory biomarkers associated with unfavorable outcomes in aSAH patients and compared their predictive ability on the prognosis. We found that almost all abnormal biomarkers related to a higher incidence of pneumonia during hospitalization, which may be an important reason for the poor prognosis of aSAH patients. Exploring the relationship between these inflammatory biomarkers and prognosis can better help clinicians understand the pathophysiological processes of aSAH and further make more active adjustments to treatment strategies.

aSAH has been shown as a state of systemic inflammation and immunosuppression [24]. It has been reported that inflammatory responses can impact the prognosis of aSAH. Still, the causes of poor prognosis are inconclusive, even though many studies have suggested that inflammation may be related to DCI, leading to unfavorable outcomes [25]. After aSAH, peripheral WBCs have been shown to migrate to the brain with activated NEUT (the most abundant type of WBC found in the circulation), causing damage to the brain and leading to a poor prognosis [11,12,26,27]. The results of this finding are in line with our previous research, showing that the WBC count remains stable in predicting 90-day outcomes [28]. In our study, WBC and NEUT both showed better predictive abilities on the prognosis of aSAH patients. To date, only a few articles focused on the MONO after aSAH. Thus, there is a very limited understanding of the role of MONO in the pathogenesis of aSAH. The possible reason is that MONO rapidly differentiates into macrophages after entering the tissue from circulation, while existing resources cannot distinguish this process. MONO can promote the inflammatory process by interacting with PLT and endothelial cells [29]. One study indicated that MONO benefits by removing subarachnoid space debris [30]. However, the other two studies showed contradictory conclusions, indicating that MONO mediates the development of cerebral vasospasm and leads to a poor prognosis [13,31]. The elevated MONO count revealed a higher incidence of in-hospital intracranial infection in our research.

Besides WBC, NEUT, and MONO, inflammation is also involved in PLT activation and thrombosis, which can concomitantly release inflammatory molecules [32]. Thus, many inflammatory biomarkers are related to PLT count [32,33]. However, there was little significant difference in PLT count between patients with favorable and unfavorable outcomes in our study. LY plays an important role in the anti-inflammatory response, but research on LY in aSAH patients is very limited [34]. One previous study indicated that T-lymphocyte inhibitors had neuroprotective effects in experimental SAH but they lacked clinical evidence [35]. In our study, there was no relationship between LY count and outcome.

In addition to the individual biomarkers such as WBC, NEUT, MONO, PLT, and LY, researchers have created some ratio-related inflammatory biomarkers to minimize the potential confounders caused by the alternation of individual blood parameters of patients. Among these inflammatory biomarkers, NLR, NAR, LMR, MLR, PWR, PNR, PLR, MPV/PLT, SIRI, and SII were associated with patient outcomes [14,15,16,17,18,19,20,21]. However, aSAH is an emergency, thus requiring clinicians to have the ability to judge the condition and make treatment decisions quickly. Among the numerous inflammatory biomarkers associated with prognosis, it is of great clinical significance to help clinicians quickly find the best indicators that may be associated with prognosis. Our study demonstrated that these biomarkers were all associated with the initial clinical status of the patient. WBC has the strongest predictive ability. Similar to almost all the other biomarkers, an elevated WBC count indicated a higher incidence of in-hospital pneumonia, which may be responsible for the poor outcomes. Our conclusion is consistent with another study, which revealed that elevated NLR was associated with higher in-hospital pneumonia after adjusting for patient baseline characteristics [14].

Unlike previous studies, which have indicated that inflammatory responses are associated with DCI [15,20], we found no relationship between abnormal inflammatory biomarkers and DCI. We speculated that although the investigators were assuming a connection between inflammation and DCI, there is currently a lack of relevant clinical evidence. The combination of complications during the patient’s hospitalization, the collection method of inflammatory biomarkers, and the confounding factors of the patient may have led the investigators to draw different conclusions.

Our previous study found that pneumonia and DCI during hospitalization may have a long-lasting impact on the prognosis of aSAH patients treated with surgical clipping and endovascular coiling [36]. It is noteworthy that nursing will become more difficult when pneumonia occurs, leading to patients staying in bed longer and increasing the risk of other severe complications. Moreover, similar to DCI, severe pneumonia would endanger the patient’s life. The clinical evidence provided by our study highlights the importance of controlling inflammation and pneumonia to improve patient outcomes.

There are various limitations of our study. First, the statistical data were retrospectively collected. Second, this study did not include some inflammatory biomarkers due to insufficient technical equipment. Third, whether these inflammatory biomarkers differ between races is unclear, so the optimal cutoff value for predicting the outcome is only of reference significance. Fourth, the AUC of 0.7 is fair and would be below the acceptance threshold for “good” predictive ability. Thus, the cutoff value of some biomarkers may not be likely to be clinically useful. Fifth, it is unclear if the patient had an infection before the onset. Sixth, not all patient biomarkers were observed at the three-time points, and the analysis is likely incomplete due to missing data. We integrated three days into one time point to use our clinical data and increase generality. We found 10, 7, and 12 inflammatory biomarkers differed across the 90-day unfavorable outcomes on Day 1, Day 2, and Day 3, respectively. Finally, subgroup-stratified analyses showed similar patterns of association as with the pooled (non-stratified) analysis in some inflammatory biomarkers groups, while others were not. The interpretation of this inconsistent finding warrants further investigations.

## 5. Conclusions

In this large retrospective study, we compared the relationship between different admission inflammatory markers and the prognosis of patients with aSAH. We found that they were all related to the clinical status of patients at admission. WBCs had the strongest association with poor functional outcomes in aSAH patients. Similar to all other inflammatory biomarkers, except monocytes, the occurrence of abnormal levels suggested the incidence of pneumonia during hospitalization, which may be an important reason for the poor outcomes of patients with aSAH. The clinical evidence provided by our study highlights the importance of controlling inflammation and pneumonia to improve patient outcomes.

## Figures and Tables

**Figure 1 brainsci-13-00257-f001:**
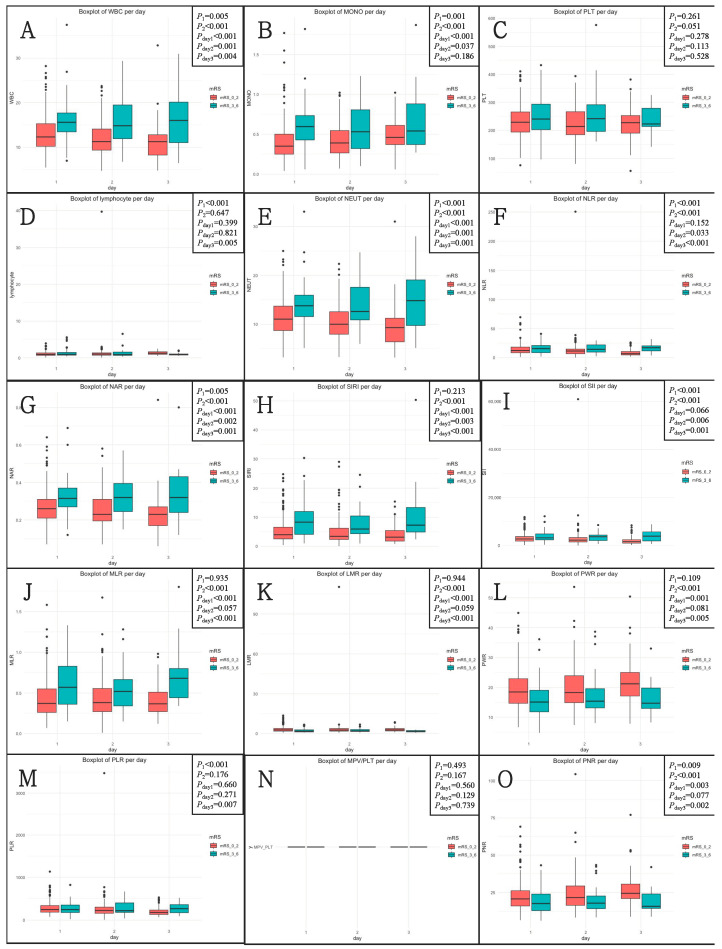
The levels of inflammatory biomarkers according to three blood sample drawn timings (Day 1, Day 2, and Day 3) across 90-day functional outcomes. (**A**) WBC; (**B**) MONO; (**C**) PLT; (**D**) lymphocyte; (**E**) NEUT; (**F**) NLR; (**G**) NLR; (**H**) SIRI; (**I**) SII; (**J**) MLR; (**K**) LMR; (**L**) PWR; (**M**) PLR; (**N**) MPV/PLT; (**O**) PNR. mRS, modified Rankin scale; NLR, neutrophil-to-lymphocyte ratio; NAR, neutrophil-to-albumin ratio; SIRI, systemic inflammation response index; SII, systemic immune-inflammation index; MLR, monocyte-to-lymphocyte ratio; LMR, lymphocyte-to-monocyte ratio; PWR, platelet-to-white blood cell ratio; PLR, platelet-to-lymphocyte ratio; MPV/PLT, mean platelet volume-to-platelet count ratio; PNR, platelet-to-neutrophil ratio.

**Figure 2 brainsci-13-00257-f002:**
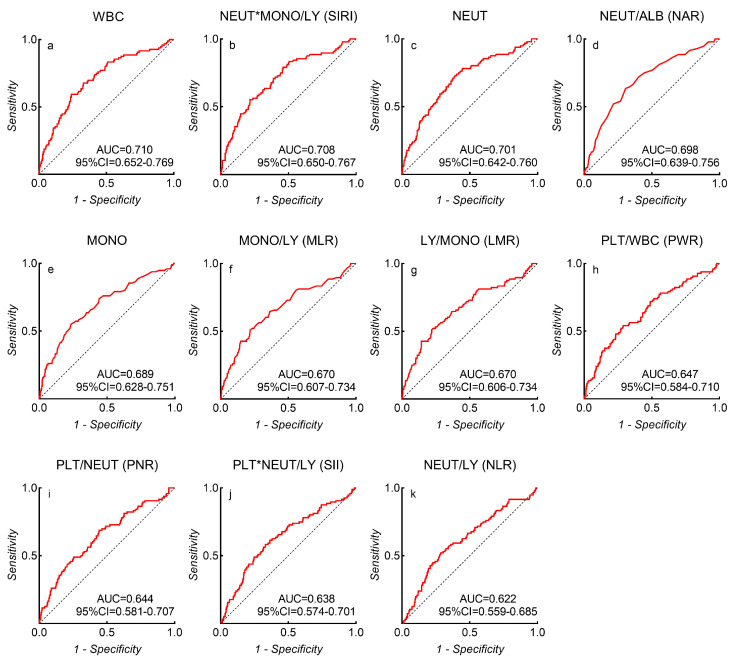
The AUC values of each inflammatory biomarker-related risk factor that predict 90-day unfavorable outcomes in aSAH patients. (**a**) WBC; (**b**) SIRI; (**c**) NEUT; (**d**) NAR; (**e**) MONO; (**f**) MLR; (**g**) LMR; (**h**) PWR; (**i**) PNR; (**j**) SII; (**k**) NLR.

**Table 1 brainsci-13-00257-t001:** Baseline characteristics.

Patient Characteristics	mRS Score at 90 Days		*p*
	mRS 0–2	mRS 3–6	
No. of patients	447	96	
Female, *n* (%)	256 (57.3)	51 (53.1)	0.457
Age, years, mean ± SD	53.9 ± 11.0	58.7 ± 10.8	<0.001
Current smoking, *n* (%)	134 (30.0)	27 (28.1)	0.718
Hypertension, *n* (%)	255 (57.0)	71 (74.0)	0.002
Hyperlipidemia, *n* (%)	40 (8.9)	7 (7.3)	0.600
Diabetes mellitus, *n* (%)	35 (7.8)	10 (10.4)	0.404
Posterior circulation, *n* (%)	53 (11.9)	11 (11.5)	0.913
WFNS grade 4–5, *n* (%)	73 (16.3)	56 (58.3)	<0.001
mFS grade 3–4, *n* (%)	339 (75.8)	89 (92.7)	<0.001
Graeb score 5–12, *n* (%)	26 (5.8)	22 (22.9)	<0.001
SEBES score 3–4, *n* (%)	213 (47.7)	50 (52.1)	0.430
Acute hydrocephalus, *n* (%)	175 (39.1)	50 (52.1)	0.020
White blood cell count *, median (IQR)	11.86 (9.40–14.68)	15.26 (12.44–18.54)	<0.001
Monocyte count *, median (IQR)	0.39 (0.26–0.54)	0.57 (0.40–0.81)	<0.001
Platelet count *, median (IQR)	225.0 (188.0–265.0)	237.5 (202.3–288.5)	0.051
Lymphocyte count *, median (IQR)	0.99 (0.70–1.38)	0.88 (0.71–1.40)	0.647
Neutrophil count *, median (IQR)	10.51 (7.97–13.27)	13.56 (11.03–16.34)	<0.001
NLR, median (IQR)	11.06 (6.97–15.70)	15.21 (8.99–21.00)	<0.001
NAR, median (IQR)	0.25 (0.19–0.31)	0.32 (0.25–0.39)	<0.001
SIRI, median (IQR)	3.68 (2.36–6.24)	7.06 (4.13–11.70)	<0.001
SII, median (IQR)	2470 (1535–3548)	3412 (2089–5002)	<0.001
MLR, median (IQR)	0.37 (0.26–0.54)	0.57 (0.36–0.79)	<0.001
LMR, median (IQR)	2.70 (1.87–3.79)	1.77 (1.26–2.79)	<0.001
PWR, median (IQR)	19.00 (14.96–23.51)	15.26 (12.13–19.80)	<0.001
PLR, median (IQR)	225.4 (162.1–313.9)	245.4 (171.5–374.9)	0.176
MPV/PLT, median (IQR)	0.04 (0.03–0.05)	0.04 (0.03–0.05)	0.167
PNR, median (IQR)	21.69 (16.60–27.92)	17.67 (13.29–21.10)	<0.001
Treatment modality			0.040
Surgical clipping, *n* (%)	200 (44.7)	54 (56.3)	
Endovascular coiling, *n* (%)	247 (55.3)	42 (43.8)	

Abbreviations: mRS, modified Rankin scale; SD, standard deviation; WFNS, world federation of neurological societies; mFS, modified fisher scale; SEBES, subarachnoid hemorrhage early brain edema; NLR, neutrophil-to-lymphocyte ratio; NAR, neutrophil-to-albumin ratio; SIRI, systemic inflammation response index; SII, systemic immune-inflammation index; MLR, monocyte-to-lymphocyte ratio; LMR, lymphocyte-to-monocyte ratio; PWR, platelet-to-white blood cell ratio; PLR, platelet-to-lymphocyte ratio; MPV/PLT, mean platelet volume-to-platelet count ratio; PNR, platelet-to-neutrophil ratio. * Unit of measurement: 10^9^/L.

**Table 2 brainsci-13-00257-t002:** Multivariate analysis results of inflammatory biomarkers after adjusting for age, hypertension, WFNS grade 4–5, mFS grade 3–4, a Graeb score of 5–12, acute hydrocephalus, and treatment modality.

Variables	OR * (95% CI)	*p*
WBC	1.15 (1.08–1.22)	<0.001
SIRI	1.09 (1.04–1.14)	<0.001
NEUT	1.14 (1.08–1.22)	<0.001
NAR	110.19 (9.42–1288.36)	<0.001
MONO	7.38 (2.75–19.76)	<0.001
MLR	4.70 (1.92–11.48)	0.001
LMR	-	-
PWR	-	-
PNR	-	-
SII	-	-
NLR	-	-

Abbreviations: OR, odds ratio; CI, confidence interval; WBC, white blood cell; SIRI, systemic inflammation response index; NEUT, neutrophil; NAR, neutrophil-to-albumin ratio; MONO, monocyte; MLR, monocyte-to-lymphocyte ratio; LMR, lymphocyte-to-monocyte ratio; PWR, platelet-to-white blood cell ratio; PNR, platelet-to-neutrophil ratio; SII, systemic immune-inflammation index; NLR, neutrophil-to-lymphocyte ratio. -, not significant. *: each multivariate regression model contains only one inflammation biomarker.

**Table 3 brainsci-13-00257-t003:** WFNS grade subgroup analysis of 90-day unfavorable outcomes.

Inflammatory Biomarkers	Subgroup	OR * (95% CI)	*p*	*p* for Interaction
WBC	WFNS 1–3	1.13 (1.04–1.22)	0.003	<0.001
	WFNS 4–5	1.17 (1.07–1.27)	<0.001	
SIRI	WFNS 1–3	1.11 (1.04–1.20)	0.003	0.697
	WFNS 4–5	1.07 (1.01–1.23)	0.016	
NEUT	WFNS 1–3	1.13 (1.04–1.22)	0.006	<0.001
	WFNS 4–5	1.16 (1.06–1.27)	0.001	
NAR	WFNS 1–3	63.17 (2.28–1752.82)	0.014	<0.001
	WFNS 4–5	71.25 (2.26–2249.10)	0.015	
MONO	WFNS 1–3	11.65 (2.70–50.25)	0.001	<0.001
	WFNS 4–5	5.41 (1.66–17.64)	0.005	
MLR	WFNS 1–3	8.94 (2.37–33.75)	0.001	0.505
	WFNS 4–5	3.55 (1.15–10.93)	0.028	
LMR	WFNS 1–3	-	-	0.074
	WFNS 4–5	-	-	
PWR	WFNS 1–3	-	-	0.063
	WFNS 4–5	-	-	
PNR	WFNS 1–3	-	-	0.110
	WFNS 4–5	-	-	
SII	WFNS 1–3	1.00 (1.00–1.00)	0.011	0.527
	WFNS 4–5	-	-	
NLR	WFNS 1–3	-	-	0.806
	WFNS 4–5	-	-	

Abbreviations: OR, odds ratio; CI, confidence interval; WBC, white blood cell; SIRI, systemic inflammation response index; NEUT, neutrophil; NAR, neutrophil-to-albumin ratio; MONO, monocyte; MLR, monocyte-to-lymphocyte ratio; LMR, lymphocyte-to-monocyte ratio; PWR, platelet-to-white blood cell ratio; PNR, platelet-to-neutrophil ratio; SII, systemic immune-inflammation index; NLR, neutrophil-to-lymphocyte ratio. -, not significant. *: each multivariate regression model contains only one inflammation biomarker.

**Table 4 brainsci-13-00257-t004:** Treatment mortality subgroup analysis of 90-day unfavorable outcomes.

Inflammatory Biomarkers	Subgroup	OR * (95% CI)	*p*	*p* for Interaction
WBC	Surgical clipping	1.18 (1.08–1.27)	<0.001	0.012
	Endovascular coiling	1.13 (1.04–1.23)	0.006	
SIRI	Surgical clipping	1.10 (1.04–1.17)	0.001	0.020
	Endovascular coiling	-	-	
NEUT	Surgical clipping	1.17 (1.08–1.26)	<0.001	0.014
	Endovascular coiling	1.13 (1.03–1.24)	0.011	
NAR	Surgical clipping	855.57 (23.85–30,698.28)	<0.001	0.005
	Endovascular coiling	-	-	
MONO	Surgical clipping	17.46 (4.22–72.19)	<0.001	0.006
	Endovascular coiling	-	-	
MLR	Surgical clipping	7.55 (2.29–24.97)	0.001	0.009
	Endovascular coiling	-	-	
LMR	Surgical clipping	0.72 (0.55–0.93)	0.013	0.258
	Endovascular coiling	-	-	
PWR	Surgical clipping	-	-	0.304
	Endovascular coiling	-	-	
PNR	Surgical clipping	-	-	0.234
	Endovascular coiling	-	-	
SII	Surgical clipping	1.00 (1.00–1.00)	0.018	0.358
	Endovascular coiling	-	-	
NLR	Surgical clipping	-	-	0.751
	Endovascular coiling	-	-	

Abbreviations: OR, odds ratio; CI, confidence interval; WBC, white blood cell; SIRI, systemic inflammation response index; NEUT, neutrophil; NAR, neutrophil-to-albumin ratio; MONO, monocyte; MLR, monocyte-to-lymphocyte ratio; LMR, lymphocyte-to-monocyte ratio; PWR, platelet-to-white blood cell ratio; PNR, platelet-to-neutrophil ratio; SII, systemic immune-inflammation index; NLR, neutrophil-to-lymphocyte ratio. -, not significant. *: each multivariate regression model contains only one inflammation biomarker.

**Table 5 brainsci-13-00257-t005:** Associations between inflammatory biomarkers and WFNS grade, mFS grade, and Graeb score on admission.

Inflammatory Biomarkers	WFNS			mFS			Graeb		
	WFNS 1–3	WFNS 4–5	*p*	mFS 1–2	mFS 3–4	*p*	Graeb 0–4	Graeb 5–12	*p*
WBC *, median (IQR)	11.70 (9.35–14.33)	15.54 (12.10–18.77)	<0.001	10.82 (8.35–12.94)	12.79 (10.46–15.94)	<0.001	12.12 (9.55–15.17)	15.87 (13.23–18.93)	<0.001
SIRI, median (IQR)	3.64 (2.32–5.92)	6.71 (3.60–12.61)	<0.001	3.04 (1.78–4.52)	4.76 (2.80–8.35)	<0.001	3.95 (2.45–6.71)	8.38 (4.27–13.28)	<0.001
NEUT *, median (IQR)	10.27 (7.70–12.74)	13.53 (10.57–16.92)	<0.001	9.12 (6.73–11.43)	11.39 (8.91–14.37)	<0.001	10.68 (8.08–13.47)	14.37 (11.76–16.74)	<0.001
NAR, median (IQR)	0.24 (0.19–0.30)	0.31 (0.25–0.39)	<0.001	0.22 (0.16–0.27)	0.27 (0.21–0.34)	<0.001	0.25 (0.19–0.31)	0.32 (0.28–0.40)	<0.001
MONO *, median (IQR)	0.39 (0.26–0.54)	0.53 (0.34–0.75)	<0.001	0.40 (0.26–0.54)	0.41 (0.28–0.59)	0.128	0.39 (0.27–0.57)	0.54 (0.40–0.89)	<0.001
MLR, median (IQR)	0.37 (0.26–0.53)	0.53 (0.35–0.76)	<0.001	0.33 (0.24–0.44)	0.42 (0.29–0.62)	<0.001	0.38 (0.27–0.56)	0.63 (0.37–0.86)	<0.001
LMR, median (IQR)	2.71 (1.89–3.83)	1.88 (1.31–2.93)	<0.001	3.00 (2.25–4.11)	2.40 (1.61–3.46)	<0.001	2.61 (1.78–3.73)	1.59 (1.17–2.70)	<0.001
PWR, median (IQR)	19.40 (15.53–23.72)	14.69 (12.14–19.10)	<0.001	22.02 (16.58–26.69)	17.37 (13.88–22.40)	<0.001	18.61 (14.79–23.43)	14.07 (9.53–19.20)	<0.001
PNR, median (IQR)	22.33 (17.44–28.84)	16.59 (13.35–20.95)	<0.001	25.71 (20.31–31.52)	19.43 (15.46–25.94)	<0.001	21.36 (16.29–27.62)	15.29 (10.63–21.17)	<0.001
SII, median (IQR)	2478 (1528–3588)	3130 (1995–5072)	<0.001	1972 (1175–3052)	2733 (1754–4156)	<0.001	2524 (1609–3827)	3145 (2218–4232)	0.039
NLR, median (IQR)	10.92 (6.91–15.68)	13.27 (8.88–22.44)	<0.001	8.33 (5.13–12.94)	12.18 (8.14–18.66)	<0.001	11.46 (7.28–16.71)	14.78 (8.89–20.86)	0.010

Abbreviations: WFNS, world federation of neurological societies; mFS, modified fisher grade; WBC, white blood cell; SIRI, systemic inflammation response index; NEUT, neutrophil; NAR, neutrophil-to-albumin ratio; MONO, monocyte; MLR, monocyte-to-lymphocyte ratio; LMR, lymphocyte-to-monocyte ratio; PWR, platelet-to-white blood cell ratio; PNR, platelet-to-neutrophil ratio; SII, systemic immune-inflammation index; NLR, neutrophil-to-lymphocyte ratio. * Unit of measurement: 10^9^/L.

**Table 6 brainsci-13-00257-t006:** Patient characteristics and group comparisons before and after propensity score matching.

Patient Characteristics	Before Propensity Score Matching			After Propensity Score Matching		
	mRS 0–2	mRS 3–6	*p*	mRS 0–2	mRS 3–6	*p*
No. of patients	447	96		86	86	
Female, *n* (%)	256 (57.3)	51 (53.1)	0.457	50 (58.1)	43 (50.0)	0.284
Age, years, mean ± SD	53.9 ± 11.0	58.7 ± 10.8	<0.001	57.8 ± 10.6	58.0 ± 10.9	0.898
Current smoking, *n* (%)	134 (30.0)	27 (28.1)	0.718	23 (26.7)	27 (31.4)	0.502
Hypertension, *n* (%)	255 (57.0)	71 (74.0)	0.002	68 (79.1)	62 (72.1)	0.287
Hyperlipidemia, *n* (%)	40 (8.9)	7 (7.3)	0.600	7 (8.1)	5 (5.8)	0.549
Diabetes mellitus, *n* (%)	35 (7.8)	10 (10.4)	0.404	7 (8.1)	9 (10.5)	0.600
Posterior circulation, *n* (%)	53 (11.9)	11 (11.5)	0.913	14 (16.3)	10 (11.6)	0.379
WFNS grade 4–5, *n* (%)	73 (16.3)	56 (58.3)	<0.001	45 (52.3)	46 (53.5)	0.879
mFS grade 3–4, *n* (%)	339 (75.8)	89 (92.7)	<0.001	83 (96.5)	79 (91.9)	0.192
Graeb score 5–12, *n* (%)	26 (5.8)	22 (22.9)	<0.001	13 (15.1)	16 (18.6)	0.541
SEBES score 3–4, *n* (%)	213 (47.7)	50 (52.1)	0.430	41 (47.7)	45 (52.3)	0.542
Acute hydrocephalus, *n* (%)	175 (39.1)	50 (52.1)	0.020	47 (54.7)	44 (51.2)	0.647
Surgical clipping, *n* (%)	200 (44.7)	54 (56.3)	0.040	43 (50.0)	48 (55.8)	0.445

Abbreviations: mRS, modified Rankin scale; SD, standard deviation; WFNS, world federation of neurological societies; mFS, modified fisher grade; SEBES, subarachnoid hemorrhage early brain edema.

**Table 7 brainsci-13-00257-t007:** Associations between inflammatory biomarkers and in-hospital complications.

**Variables**	**WBC ***			**SIRI**			**NEUT ***		
	**>14.82**	**≤14.82**	** *p* **	**>6.77**	**≤6.77**	** *p* **	**>11.39**	**≤11.39**	** *p* **
	**N = 80**	**N = 92**		**N = 74**	**N = 98**		**N = 100**	**N = 72**	
Delayed cerebral ischemia, *n* (%)	36 (45.0)	34 (37.0)	0.284	33 (44.6)	37 (37.8)	0.366	42 (42.0)	28 (38.9)	0.682
Intracranial infection, *n* (%)	12 (15.0)	10 (10.9)	0.419	8 (10.8)	14 (14.3)	0.499	15 (15.0)	7 (9.7)	0.307
Stress ulcer bleeding, *n* (%)	19 (23.8)	29 (31.5)	0.257	22 (29.7)	26 (26.5)	0.643	24 (24.0)	24 (33.3)	0.178
Hypoproteinemia, *n* (%)	40 (50.0)	38 (41.3)	0.253	35 (47.3)	43 (43.9)	0.656	47 (47.0)	31 (43.1)	0.608
Pneumonia, *n* (%)	55 (68.8)	38 (41.3)	<0.001	49 (66.2)	44 (44.9)	0.006	63 (63.0)	30 (41.7)	0.006
Deep vein thrombosis, *n* (%)	13 (16.3)	12 (13.0)	0.552	14 (18.9)	11 (11.2)	0.156	16 (16.0)	9 (12.5)	0.521
**Variables**	**NAR**			**MONO ***			**MLR**		
	**>0.29**	**≤0.29**	** *p* **	**>0.55**	**≤0.55**	** *p* **	**>0.56**	**≤0.56**	** *p* **
	**N = 86**	**N = 86**		**N = 67**	**N = 105**		**N = 69**	**N = 103**	
Delayed cerebral ischemia, *n* (%)	37 (43.0)	33 (38.4)	0.535	28 (41.8)	42 (40.0)	0.816	28 (40.6)	42 (40.8)	0.979
Intracranial infection, *n* (%)	13 (15.1)	9 (10.5)	0.361	13 (19.4)	9 (8.6)	0.038	8 (11.6)	14 (13.6)	0.701
Stress ulcer bleeding, *n* (%)	23 (26.7)	25 (29.1)	0.734	20 (29.9)	28 (26.7)	0.650	20 (29.0)	28 (27.2)	0.796
Hypoproteinemia, *n* (%)	43 (50.0)	35 (40.7)	0.221	30 (44.8)	48 (45.7)	0.904	31 (44.9)	47 (45.6)	0.928
Pneumonia, *n* (%)	59 (68.6)	34 (39.5)	<0.001	40 (59.7)	53 (50.5)	0.236	45 (65.2)	48 (46.6)	0.016
Deep vein thrombosis, *n* (%)	14 (16.3)	11 (12.8)	0.516	13 (19.4)	12 (11.4)	0.148	12 (17.4)	13 (12.6)	0.384
**Variables**	**LMR**			**PWR**			**PNR**		
	**<1.79**	**≥1.79**	** *p* **	**<15.62**	**≥15.62**	** *p* **	**<20.72**	**≥20.72**	** *p* **
	**N = 71**	**N = 101**		**N = 80**	**N = 92**		**N = 102**	**N = 70**	
Delayed cerebral ischemia, *n* (%)	30 (42.3)	40 (39.6)	0.728	33 (41.3)	37 (40.2)	0.891	40 (39.2)	30 (42.9)	0.633
Intracranial infection, *n* (%)	8 (11.3)	14 (13.9)	0.616	12 (15.0)	10 (10.9)	0.419	15 (14.7)	7 (10.0)	0.364
Stress ulcer bleeding, *n* (%)	20 (28.2)	28 (27.7)	0.949	22 (27.5)	26 (28.3)	0.912	30 (29.4)	18 (25.7)	0.595
Hypoproteinemia, *n* (%)	33 (46.5)	45 (44.6)	0.803	38 (47.5)	40 (43.5)	0.597	50 (49.0)	28 (40.0)	0.243
Pneumonia, *n* (%)	47 (66.2)	46 (45.5)	0.008	53 (66.3)	40 (43.5)	0.003	64 (62.7)	29 (41.4)	0.006
Deep vein thrombosis, *n* (%)	12 (16.9)	13 (12.9)	0.460	14 (17.5)	11 (12.0)	0.304	19 (18.6)	6 (8.6)	0.066
**Variables**	**SII**			**NLR**					
	**>3102**	**≤3102**	** *p* **	**>14.88**	**≤14.88**	** *p* **			
	**N = 84**	**N = 88**		**N = 74**	**N = 98**				
Delayed cerebral ischemia, *n* (%)	36 (42.9)	34 (38.6)	0.573	31 (41.9)	39 (39.8)	0.782			
Intracranial infection, *n* (%)	13 (15.5)	9 (10.2)	0.303	12 (16.2)	10 (10.2)	0.243			
Stress ulcer bleeding, *n* (%)	22 (26.2)	26 (29.5)	0.624	18 (24.3)	30 (30.6)	0.363			
Hypoproteinemia, *n* (%)	43 (51.2)	35 (39.8)	0.133	37 (50.0)	41 (41.8)	0.287			
Pneumonia, *n* (%)	55 (65.5)	38 (43.2)	0.003	47 (63.5)	46 (46.9)	0.031			
Deep vein thrombosis, *n* (%)	14 (16.7)	11 (12.5)	0.438	14 (18.9)	11 (11.2)	0.156			

Abbreviations: WBC, white blood cell; SIRI, systemic inflammation response index; NEUT, neutrophil; NAR, neutrophil-to-albumin ratio; MONO, monocyte; MLR, monocyte-to-lymphocyte ratio; LMR, lymphocyte-to-monocyte ratio; PWR, platelet-to-white blood cell ratio; PNR, platelet-to-neutrophil ratio; SII, systemic immune-inflammation index; NLR, neutrophil-to-lymphocyte ratio. * Unit of measurement: 10^9^/L.

## Data Availability

The data supporting the findings of this study are available from the corresponding author upon reasonable request.

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
