# Peer review of "A Pooled Analysis of Preoperative Inflammatory Biomarkers to Predict 90-Day Outcomes in Patients with an Aneurysmal Subarachnoid Hemorrhage: A Single-Center Retrospective Study"

_brainsci, 2023, doi:10.3390/brainsci13020257_

Round 1
Reviewer 1 Report
I have some questions about the manuscript which described the inflammatory biomarkers to predict 90-day outcomes in patients with aneurysmal subarachnoid hemorrhage (SAH).
Regarding the design of the study, I wonder why the authors have almost exclusively focused on the inflammatory biomarkers instead of more critical factors in case of SAH patients.
As the authors mention, the most important cause of the poor prognosis is generally known to the degree of the primary damage of the brain due to the enormous rapid increase of intracranial pressure by SAH. In general, prognosis of the patients with SAH was mostly dependent on the H&K and/ or WFNS grade.
If the authors want to find out the inflammatory markers related to the poor prognosis of patients with SAH, Why don’t they analysis the inflammatory markers between the favorable and unfavorable group in the same H&K and/ or WFNS grade patients.
Another critical concern is that the data was based on retrospective study instead of prospective study.
Author Response
Title of the Manuscript: A pooled analysis of preoperative inflammatory biomarkers to predict 90-day outcomes in patients with aneurysmal sub-arachnoid hemorrhage: a single-center retrospective study
Manuscript Number: brainsci-2108244
We thank the Reviewers for their positive comments and constructive suggestions. We have revised the manuscript accordingly. We carefully quality‐checked our manuscript, and revisions reviewers mentioned or not to the manuscript are tracked.
Reviewer 1
Comment 1
I have some questions about the manuscript which described the inflammatory biomarkers to predict 90-day outcomes in patients with aneurysmal subarachnoid hemorrhage (SAH).
Regarding the design of the study, I wonder why the authors have almost exclusively focused on the inflammatory biomarkers instead of more critical factors in case of SAH patients. As the authors mention, the most important cause of the poor prognosis is generally known to the degree of the primary damage of the brain due to the enormous rapid increase of intracranial pressure by SAH. In general, prognosis of the patients with SAH was mostly dependent on the H&K and/ or WFNS grade. If the authors want to find out the inflammatory markers related to the poor prognosis of patients with SAH, why don’t they analysis the inflammatory markers between the favorable and unfavorable group in the same H&K and/ or WFNS grade patients.
Author's Response
Thank you very much for your valuable comments. Although inflammatory biomarkers all perform well in predicting poor outcomes according to the previous studies, unfortunately, most of these biomarkers have been reported alone and adjusted for different critical factors. The objectives of our present study were to analyze the associations between inflammatory biomarkers and outcomes, to compare the ability of these inflammatory biomarkers to predict outcomes and to find out potential reasons for these inflammatory biomarkers leading to unfavorable outcomes. In table 2, each multivariate analysis result of inflammatory biomarkers was adjusted for critical factors in the fields of aSAH, such as age, hypertension, WFNS grade 4-5, mFS grade 3-4, Graeb 5-12, acute hydrocephalus, and treatment modality. After adjusting for the clinical variables, the multivariate analysis showed that WBC count, SIRI, NEUT count, NAR, MONO count, and MLR were still independently associated with 90-day unfavorable outcomes in aSAH patients.
Change to Text
None.
Comment 2
Another critical concern is that the data was based on retrospective study instead of prospective study.
Author's Response
Thank you very much for your valuable comments. Indeed, this is a retrospective study instead of a prospective design. All patients’ data were derived from the Long-term Prognosis of Emergency Aneurysmal Subarachnoid Hemorrhage (LongTEAM) Registry study (Registration No. NCT04785976). This observational clinical trial improves the diagnosis and treatment effect and efficiency in this field, reducing mortality, medical costs, and medical burden while opening up new avenues for interdisciplinary clinical practice and scientific research exploration. The present study was one little step to keeping our mind on originality.
Change to Text
None.

Reviewer 2 Report
Thank you for your efforts!
- Language editing is needed.
- Important recent references are missing:
PMID: 35927663
PMID: 35567655
Author Response
Title of the Manuscript: A pooled analysis of preoperative inflammatory biomarkers to predict 90-day outcomes in patients with aneurysmal sub-arachnoid hemorrhage: a single-center retrospective study
Manuscript Number: brainsci-2108244
We thank the Reviewers for their positive comments and constructive suggestions. We have revised the manuscript accordingly. We carefully quality‐checked our manuscript, and revisions reviewers mentioned or not to the manuscript are tracked.
Reviewer 2
Comment 1
Thank you for your efforts!
- Language editing is needed.
- Important recent references are missing:
PMID: 35927663
PMID: 35567655
Author's Response
We thank the reviewer for this kind suggestion. The language editing of the manuscript was carefully modified word by word. As suggested by the reviewer, we have added two above-mentioned references in the manuscript.
Change to Text
“The results of this finding are in line with our previous research, showing WBC count remains stable in predicting 90-day outcomes (28).”
We supplemented this sentence in lines 297-299.

Reviewer 3 Report
This is a manuscript that investigated the utility of an admission inflammatory biomarker for the prediction of outcome after aSAH. The authors found that increased admission WBC is associated with increased incidence of pneumonia and a lower mRS score at 90 days. Interestingly, contrary to prior report, the was no association with the incidence of vasospasm. The manuscript is well written and the data well presented. A very interesting aspect is the various novel inflammatory markers and ratios which have been proposed seem less useful than the simple WBC.
1) Do any of your patients receive corticosteroids?
2) Do you have data on the interval between the aSAH and admission? I imagine some of the patients may have transferred from other hospitals and your inclusion criteria is aSAH within 72hrs. The timing of the admission labs i.e. whether taken at 1hr or 72hrs after the aSAH is likely important. If the data is available you should control for time since aSAH in your analysis. If not, this should be stated as a limitation.
3) Line 294:
Among the numerous inflammatory biomarkers associated with prognosis, how helps clinicians rapidly figure out the best indicator that may be related to prognosis has important clinical significance.
Please clarify the line as it is not clear what is meant by the sentence.
Author Response
Title of the Manuscript: A pooled analysis of preoperative inflammatory biomarkers to predict 90-day outcomes in patients with aneurysmal subarachnoid hemorrhage: a single-center retrospective study
Manuscript Number: brainsci-2108244
We thank the Reviewers for their positive comments and constructive suggestions. We have revised the manuscript accordingly. We carefully quality‐checked our manuscript, and revisions reviewers mentioned or not to the manuscript are tracked.
Reviewer 3
Comment 1
This is a manuscript that investigated the utility of an admission inflammatory biomarker for the prediction of outcome after aSAH. The authors found that increased admission WBC is associated with increased incidence of pneumonia and a lower mRS score at 90 days. Interestingly, contrary to prior report, the was no association with the incidence of vasospasm. The manuscript is well written and the data well presented. A very interesting aspect is the various novel inflammatory markers and ratios which have been proposed seem less useful than the simple WBC.
1) Do any of your patients receive corticosteroids?
Authors’ Response
Thanks for your advice. We don't use corticosteroids routinely. If negative fluid balance persists in some patients, hormonal therapy, such as hydrocortisone or hydrocortisone, may be considered. It can reduce the amount of fluid needed to maintain equal blood volume and help correct hyponatremia. However, it should be used with caution or prohibited for those with gastrointestinal bleeding and increased blood sugar. If cerebral salt wasting syndrome suspected, corticosteroids will be considered, though the effect not that obvious. Another point we're just thinking about patients with cerebral edema. We use corticosteroids routinely only for severe infections such as septic shock, and abnormal preoperative inflammatory markers are not indicative. The corticosteroids application you proposed is a promising direction for us to practice.
Change to Text
None.
Comment 2
2) Do you have data on the interval between the aSAH and admission? I imagine some of the patients may have transferred from other hospitals and your inclusion criteria is aSAH within 72hrs. The timing of the admission labs i.e., whether taken at 1hr or 72hrs after the aSAH is likely important. If the data is available you should control for time since aSAH in your analysis. If not, this should be stated as a limitation.
Author's Response
Thanks for your suggestion. The time interval from aneurysm rupture to admission is available in our database.
We divided WBC count into three groups within 72 hours according to the blood sample drawn time (Day 1 [rupture to admission interval between 0-24 hours], Day 2 [rupture to admission interval between 25-48 hours], Day 3 [rupture to admission interval between 49-72 hours]). It’s very constructive advice for our article. We reviewed some previous studies and did find there is a trend or differences in the predictive ability of different blood sample drawing times (1-3). We found 10, 7, and 12 inflammatory biomarkers differed across the 90-day unfavorable outcomes on Day 1, Day 2, and Day 3, respectively. We integrated three days into a one-time point to fully use our clinical data and increase generality. It’s a limitation of our study.
Change to Text
“The blood sample drawn timing of inflammatory biomarkers was divided into three groups (Day 1 [rupture to admission interval between 0-24 hours], Day 2 [rupture to admission interval between 25-48 hours], Day 3 [rupture to admission interval between 49-72 hours]) to find out whether there are different levels in the predefined groups over time. ” We supplemented this sentence in the line 121-124.
“After the rupture event occurred, the levels of inflammatory biomarkers were an-alyzed. Two hundred ninety-eight patients (54.9%) received tests on post-hemorrhagic day 1, 158 (29.1%) on day 2, and 87 (16.0%) on day 3. The levels of inflammatory bi-omarkers according to three blood sample drawn timing were shown in Figure 1. On Day 1, Day 2, and Day 3, the levels of 10, 7, and 12 inflammatory biomarkers, respectively, were different across the 90-day unfavorable outcomes.” We supplemented this sentence in the line 162-167.
We added Figure 1 (Levels of inflammatory biomarkers across 90-day functional outcomes.) in page 6.
“Finally, we found 10, 7, and 12 inflammatory biomarkers differed across the 90-day unfavorable outcomes on Day 1, Day 2, and Day 3, respectively. We integrated three days into a one-time point to make full use of our clinical data and increase generality. ”We supplemented this sentence in the line 351-353.
Comment 3
3) Line 294:
Among the numerous inflammatory biomarkers associated with prognosis, how helps clinicians rapidly figure out the best indicator that may be related to prognosis has important clinical significance.
Please clarify the line as it is not clear what is meant by the sentence.
Author's Response
Thanks for your detailed comment. We made revisions to the manuscript.
Change to Text
“Among the numerous inflammatory biomarkers associated with prognosis, it is of great clinical significance to help clinicians quickly find the best indicators that may be associated with prognosis.”
We supplemented this sentence in lines 324-326.

Round 2
Reviewer 1 Report
I have a simple and substantial concern about this manuscript regarding the design of statistical materials and methods.
Shown in Table 1 (Baseline characteristics), WFNS grade 4-5, n (%) represented 73/447 (16.3%) in favourable and 56/96 (58.3%) in unfavourable group, <0.001, respectively. Then, it is obvious these two groups fundamentally consist of different outcome patients.
Although the authors have performed propensity score matching (PSM) to adjust for imbalances of baseline characteristics such as age, hypertension, WFNS grade, mFS grade, Graeb score, acute hydrocephalus, and surgical clipping between the two outcome groups, there is a critical difference between the two groups, favourable and unfavourable group divided by mRS scores.
Again, I am not convinced with the study's design to compare the two groups, no matter what kind of statistical analysis was applied to detect the biomarker in these groups.
The study aims to analyze the associations between inflammatory biomarkers and outcomes of SAH patients by using multivariate analysis.
If so, at least multiple variants analysis should have been applied for the two groups in the same WFNS grade group and the same treatment group.
Author Response
Thank you for reviewing our manuscript and for the constructive comments, which greatly helped us to improve the manuscript. In this large retrospective study, we compared the relationship between different admission inflammatory markers and the 90-day unfavorable outcomes of patients with aneurysmal subarachnoid hemorrhage. Several forward stepwise likelihood ratio test multivariate logistic models to identify the inflammation-related independent risk factors associated with 90-day unfavorable outcomes were established, followed by subgroup and interaction analyses in the same manner.
After performing interaction analyses, we found that the subgroup population was partly the same as the overall population. At the same time, there are some interactions and relationships between levels of inflammatory biomarkers and WFNS grade and treatment modality, respectively.
We have revised the manuscript per the comments and marked all the amends on our revised manuscript.
